# Multiple Sclerosis Onset before and after COVID-19 Vaccination: Can HLA Haplotype Be Determinant?

**DOI:** 10.3390/ijms25084556

**Published:** 2024-04-22

**Authors:** Assunta Bianco, Gabriele Di Sante, Francesca Colò, Valeria De Arcangelis, Alessandra Cicia, Paola Del Giacomo, Maria De Bonis, Tommaso Giuseppe Morganti, Vincenzo Carlomagno, Matteo Lucchini, Angelo Minucci, Paolo Calabresi, Massimiliano Mirabella

**Affiliations:** 1Division of Neurology, Fondazione Policlinico Universitario A. Gemelli IRCCS, 00168 Rome, Italy; 2Department of Neurosciences, Centro di Ricerca per la Sclerosi Multipla “Anna Paola Batocchi”, Catholic University of Sacred Heart, 00168 Rome, Italy; 3Department of Medicine and Surgery, Section of Human, Clinical and Forensic Anatomy, University of Perugia, 06123 Perugia, Italy; 4Department of Laboratory and Infectious Sciences, Fondazione Policlinico Universitario A. Gemelli IRCCS, 00168 Rome, Italy; 5Departmental Unit of Molecular and Genomic Diagnostics, Fondazione Policlinico Universitario A. Gemelli IRCCS, 00168 Rome, Italy; 6Genomics Core Facility, Gemelli Science and Technology Park (G-STeP), Fondazione Policlinico Universitario A. Gemelli IRCCS, 00168 Rome, Italy

**Keywords:** multiple sclerosis, SARS-CoV-2 vaccination, COVID-19 and autoimmune disorders, HLA-DRB1 risk factors

## Abstract

A few cases of multiple sclerosis (MS) onset after COVID-19 vaccination have been reported, although the evidence is insufficient to establish causality. The aim of this study is to compare cases of newly diagnosed relapsing–remitting MS before and after the outbreak of the COVID-19 pandemic and the impact of COVID-19 vaccination. Potential environmental and genetic predisposing factors were also investigated, as well as clinical patterns. This is a single-centre retrospective cohort study including all patients who presented with relapsing–remitting MS onset between January 2018 and July 2022. Data on COVID-19 vaccination administration, dose, and type were collected. HLA-DRB1 genotyping was performed in three subgroups. A total of 266 patients received a new diagnosis of relapsing–remitting MS in our centre, 143 before the COVID-19 pandemic (until and including March 2020), and 123 during the COVID-19 era (from April 2020). The mean number of new MS onset cases per year was not different before and during the COVID-19 era and neither were baseline patients’ characteristics, type of onset, clinical recovery, or radiological patterns. Fourteen (11.4%) patients who subsequently received a new diagnosis of MS had a history of COVID-19 vaccination within one month before symptoms onset. Patients’ characteristics, type of onset, clinical recovery, and radiological patterns did not differ from those of patients with non-vaccine-related new diagnoses of MS. The allele frequencies of HLA-DRB1*15 were 17.6% and 22.2% in patients with non-vaccine-related disease onset before and during the COVID-19 era, respectively, while no case of HLA-DRB1*15 was identified among patients with a new diagnosis of MS post-COVID-19 vaccine. In contrast, HLA-DRB1*08+ or HLA-DRB1*10+ MS patients were present only in this subgroup. Although a causal link between COVID-19 vaccination and relapsing–remitting MS cannot be detected, it is interesting to note and speculate about the peculiarities and heterogeneities underlying disease mechanisms of MS, where the interactions of genetics and the environment could be crucial also for the follow-up and the evaluation of therapeutic options.

## 1. Introduction

Multiple sclerosis (MS) is a chronic, immune-mediated disease of the central nervous system (CNS) characterized by focal demyelination and neurodegeneration. The aetiopathogenesis of MS is multifactorial, and environmental factors have been shown to have a considerable impact on susceptible individuals [1,2,3]. Among triggers, viral infections have been associated with an increased risk of developing MS [4,5,6,7,8], and a direct, causal relation has recently been established for the Epstein–Barr virus (EBV) [9]. Active immunization with vaccines has been also hypothesized as a possible trigger for MS onset on the assumption that immune stimulation may lead to a dysregulated, auto-aggressive response. Nevertheless, the risk of developing MS after vaccinations against Hepatitis B virus, Human Papillomavirus, seasonal influenza and H1N1, measles–mumps–rubella, Variola, tetanus, Bacillus Calmette–Guérin, Poliovirus, typhoid fever, and diphtheria has not been proven [10,11,12].

The recent COVID-19 pandemic and the subsequent vaccination campaign have raised a new debate on the mutual relations between viral infections, vaccinations, and MS. The massive inflammatory response induced by SARS-CoV-2 infection is peculiar compared to other viruses [13,14] and has been associated with the occurrence of many para- and post-infectious inflammatory neurological conditions, like acute disseminated encephalomyelitis (ADEM) [15,16] and neuromyelitis optica spectrum disorders (NMOSDs) [17,18]. A few cases of typical onset of MS during or post-COVID-19 have been reported as well, although the evidence is insufficient to establish a causative role of the SARS-CoV-2 virus [19,20,21,22,23].

After the early stages of vaccination campaigns, in late 2020, many cases of neurological disorder onset after vaccination were reported [24,25,26,27]. A few cases of reactivation or new onset of demyelinating diseases of the CNS have been described as well, both after vaccination with mRNA-based [28,29,30,31] and with adenovirus-vectored formulations of the vaccine [31,32,33]. The assumption that anti-COVID-19 vaccination may increase the risk of developing MS is still debated, and this potential causal relationship could change the risk–benefit profile of anti-COVID-19 vaccination, at least in some population subgroups. Genetic and environmental predisposing factors for the onset of demyelinating disease after the vaccine in predisposed individuals have not yet been defined.

The T-cell-mediated self-reactivity against myelin is strongly linked to the human leucocyte antigen (HLA)-DRB1. In particular, the HLA-DRB1*15 haplotype represents the single genetic factor with the strongest association with relapsing–remitting MS [34,35,36], although more recent genome-wide association studies showed extreme complexity of interactions between genes and immune response in the development of MS [37,38].

This study aimed to compare cases of newly diagnosed relapsing–remitting MS reported to our MS centre before and after the outbreak of the COVID-19 pandemic and their temporal relationship with SARS-CoV-2 vaccination. Potential environmental and genetic predisposing factors were also investigated, as well as clinical patterns of the post-vaccine MS cohort and a typical newly diagnosed MS cohort.

## 2. Results

### 2.1. Comparison between Patients with a Disease before and during the COVID-19 Pandemic

During the study period (January 2018–July 2022), 266 patients received a new diagnosis of relapsing–remitting MS at our centre, 143 before the SARS-CoV-2 outbreak (until and including March 2020), and 123 during the COVID-19 era (from April 2020). The mean number of new MS onset was 64 per year before the COVID-19 era and 56 per year during the COVID-19 era; a statistically significant difference was not detectable (Figure 1). Similarly, the number of new MS onset per trimester was not statistically different before and during the COVID-19 era (Figure 2). One hundred eighty-five (69.5%) patients were female, with a median age of 34 years, and 10.8% of them had another associated autoimmune disease. The most common onset types were spinal (27.1%), multisystemic (21.4%), and brain stem (19.9%). Oligoclonal bands were positive in 83.5% of patients. Median EDSS after relapse was 1.0. Complete or partial recovery after pulse steroid treatment was observed in 93.6% of patients. MRI at onset showed a median of five supratentorial T2 lesions; 54.1% of patients showed at least one gadolinium-enhancing lesion. A comparison between patients with a new diagnosis of MS before and during the COVID-19 era showed no difference in patients’ characteristics, type of onset, clinical recovery, and radiological patterns; during the COVID-19 era, the time from onset to diagnosis was lower than that of patients diagnosed before SARS-CoV-2 advent (Table 1). None of the patients of the MS cohort with disease onset during the COVID-19 pandemic documented or experienced a SARS-CoV-2 infection within 1 month before symptoms onset.

### 2.2. Analysis of MS Cohort with a SARS-CoV-2 Vaccine-Related Disease Onset

Fourteen (11.4%) patients who subsequently received a new diagnosis of MS had a history of SARS-CoV-2 vaccination within one month before symptom onset. The demyelinating event occurred after the first dose vaccination in three patients, after the second dose vaccination in seven patients, and after booster dose vaccination in four patients (Figure 3). Ten patients received m-RNA BNT162b2 (Figure 3), three patients received mRNA-1273, and one patient received heterologous vaccination (ChAdOx1-S/mRNA-1273). Patients’ characteristics, type of onset, clinical recovery, and radiological patterns did not differ from those of patients with non-vaccine-related new diagnoses of MS (Table 2). A sub-analysis was performed to test the hypothesis of whether patients with post-vaccine onset differed in the severity of disease. No statistically significant difference was detectable in the EDSS in presentation, type of relapse, and recovery after relapse in univariate analysis post-PSM (Appendix A).

An additional four patients developed inflammatory demyelinating events within the second month after vaccine exposure (three patients after the second dose and one patient after the booster dose); three patients received heterologous vaccination (ChAdOx1-S/mRNA BNT162b2), and one patient received BNT162b2. Patients’ characteristics, type of onset, clinical recovery, and radiological patterns are shown in Appendix A and did not differ from patients with non-vaccine-related new diagnoses of MS.

### 2.3. Comparison of HLA-DRB1 Genotyping before and during COVID-19 Pandemic

HLA genotyping was performed on plasma samples collected from 49 MS patients: 17 patients with disease onset before the COVID-19 era (control group 1), 18 patients with disease onset during the COVID-19 era but not vaccine-related (control group 2), and all 14 patients with a post-vaccine new diagnosis of MS (cases). HLA-DRB1 alleles in the three groups are reported in Table 3 and Appendix A. The allele frequencies of HLA-DRB1*15 were 17.6% and 22.2% in control groups 1 and 2, respectively, but none of the patients with a new diagnosis of MS post-COVID-19 vaccine resulted in HLA-DRB1*15 at least in one of the two alleles, while HLA-DRB1*08^pos^ and HLA-DRB1*10^pos^ MS patients were present only in the subgroup of our cohort with a vaccine-related disease onset (Figure 4).

## 3. Discussion

Our study showed a different HLA genotype pattern in patients with relapsing–remitting MS onset within one month after the COVID-19 vaccine compared to both patients with non-vaccine-related MS onset and patients with MS onset before the COVID-19 pandemic. The genotype of HLA is not modifiable by immune response in any case, but, interestingly, we observed that certain HLA haplotypes recurred more frequently in patients with MS onset within one month after the COVID-19 vaccine. In this case, the genotype is a prerequisite, and probably the immune system after the vaccine in selected people with a genetic predisposition and probably with certain HLA haplotypes can enhance specific mechanisms of response, in some cases protective and others dangerous. Self-reactive mechanisms are largely involved in MS pathogenesis, where CNS-reactive T and B cells play a prominent role [39,40,41,42]. Thus, peripherally activated T cells specific for myelin antigens become able to cross the blood–brain barrier [43,44]. After reaching CNS, they recognize specific targets able to re-stimulate them through local antigen-presenting cells (APCs) [45,46,47], causing damage to myelin sheaths and recruitment of other immune cells [48,49,50,51], leading to demyelination. The T cell-mediated self-reactivity against myelin is strongly linked to the human leucocyte antigen HLA-DRB1*15 allele (HLA-DR2) [52]. Although it is well known that the HLA-DRB1*15 haplotype represents a single genetic factor with the strongest association with MS [36], especially with relapsing–remitting MS (OR = 3.08), more recent genome-wide association studies (GWASs) dissected the various and heterogeneous universe of genes associated with MS [37,38], while cell-specific fine-mapping revealed the complexity of these interactions, providing novel evidence about the potential genetic mechanisms that independently involve T and B cells in the pathogenesis of MS [53]. These findings are embedded in the long-standing debate regarding the aetiology of MS, corroborating the causal role of both CD4 T cells, including Th17 T cells, and memory B cells [3,54,55], confirmed by the highly effective therapies targeting these subpopulations [56].

In this study, we intended to compare relapsing–remitting MS cohorts based on their onset in three distinct periods: before and during the COVID-19 era, distinguishing between MS patients with demyelinating event onset within 1 month after COVID-19 vaccination and patients with non-vaccine-related MS onset. The time interval by which to define a temporal relationship between a triggering event (e.g., a vaccine) and the onset of MS is controversial. We established a one-month timeframe based on recent literature about inflammatory events related to SARS-CoV-2 vaccination [26]; the same timeframe was chosen by Alluqmani in his case–control study including thirty-two patients with vaccine-related MS onset [31].

The association between COVID-19 vaccination and MS onset is still debated in the literature [57,58]. Havla and Coll. described a case of MS onset in a 28-year-old woman six days after the first dose of the BNT162b2 mRNA vaccine [28]. Khayat-Khoei and Coll. described three cases of new-onset MS or neuromyelitis optica and four cases of exacerbation of known stable MS within 21 days after the first or second dose of the mRNA-1273 or the BNT162b2 vaccines [29]. Toljan and Coll. described a series of five patients who developed MS during 1-day and two-month timeframes after an mRNA-based COVID-19 vaccine [31]. In the Oxford COVID Vaccine Trial, one case of MS reactivation 10 days after the first dose of the ChAdOx1 vaccine was observed among 11,636 participants [32]. Rinaldi and Coll. collected three cases of MS onset occurring within 35 days after the COVID-19 vaccine and performed a systematic review that included 47 cases of MS onset/reactivation after the COVID-19 vaccine [33]. To avoid confounding, we evaluated only demyelinating events at disease onset in our study, not including cases of MS reactivation as reported in other works [33,59,60]. In our study, we were not able to find any demographic, clinical, or radiological differences between patients with post-vaccine new-onset MS and patients with non-vaccine-relate MS onset. Establishing a causal relationship between vaccination against COVID-19 and the onset of MS is outside the objectives of this study, although the mean number of new MS onset cases per year was not different before and during the COVID-19 era in our centre. To confirm or deny a causal association between vaccination against COVID-19 and MS onset, one or more case–control studies in large populations will be needed, as was the case for vaccinations against HBV and HPV [10].

The HLA-DRB1 genotyping of MS patients reflects frequencies normally expected both in line with the Italian reference population (allefrequencies.net, last visit to website: 6 June 2023) [61] and with what is generally expected in MS patient cohorts [34,35]. There is a slight non-significant prevalence of certain alleles generally associated with autoimmune diseases, and the haplotype combinations reflect those most frequent in the reference population. Intriguingly, the genotyping of HLA-DRB1, comparing the patients based on disease onset (before or during the COVID-19 pandemic and related or unrelated to SARS-CoV-2 vaccination), revealed that none of the MS patients with demyelinating event onset within one month after COVID-19 vaccination expressed the HLA-DRB1*15 allele. In contrast, the HLA-DRB1*08 and HLA-DRB1*10 alleles were expressed in the vaccine-related onset cohort but not in the other two groups of MS patients with disease onset unrelated to either SARS-CoV-2 infection or vaccination.

Although these observations regard a small number of individuals, these results are intriguing and open to distinct speculations. First, the HLA-DRB1*15 allele generally averages around one-fifth of MS Caucasian Italian patients [34,35,56]. The fact that no patient in this group presented the single genetic risk factor related to MS and was able to evoke a mechanism mediated by abnormal antigen presentation and autoimmune T lymphocyte recruitment processes may suggest that other potential mechanisms are predominant, probably triggered by the vaccination itself or other bystander effects following the vaccination. This paper does not intend to find a causal link between COVID-19 vaccination and MS, in line with the literature where this topic remains still controversial [28,29,30,31,59,60]. Heterogeneous humoral and cellular immune responses to the different doses and formulations of COVID-19 vaccination have been widely reported [62,63]. In line with this, the alleles exclusively expressed in the vaccine-related cohort, although rarely and poorly associated with immune-related disorders, have been described to be predisposing risk factors for MS and other autoimmune diseases (HLA-DRB1*08 [64,65]) and IgG4 mediated encephalopathies and neuromyelitis optica (HLA-DRB1*10 [66,67,68,69]). The fact that HLA-DRB1*08^pos^ and HLA-DRB1*10^pos^ MS patients are limited to the cohort of those with a disease onset related to SARS-CoV-2 vaccination suggests a possible B cell or antibodies-mediated mechanism(s) probably responsible for initiating or triggering the disease in individuals predisposed to or at risk of developing MS.

Nevertheless, it will be interesting to follow up on the evolution of the disease in these patients and to understand whether they will develop a more severe course over time or be amenable to more targeted therapies. For example, we can speculate that they might benefit from an approach more directed at the B cell compartment, being prevalent T cell-independent mechanisms in their pathogenesis.

Our study has several limitations. It was not possible to analyse the HLA pattern for all patients included in the study. However, we analysed all patients with a post-COVID-19 vaccine new diagnosis of MS and an equivalent number of patients in the two control groups. Our study did not include a healthy individual control group. However, since we enrolled only non-Sardinian Italian Caucasian patients, we used the data available on the general Italian population as a reference for the frequencies of the HLA haplotypes. Furthermore, being a single-centre study, its generalizability requires further confirmation.

A detailed discussion of the efficacy and safety of COVID-19 vaccination in patients who already have a diagnosis of MS (whether or not being treated with disease-modifying drugs) is outside the objectives of this study.

## 4. Materials and Methods

### 4.1. Study Population

This is a single-centre retrospective cohort study conducted at the MS Centre of Fondazione Policlinico Universitario Agostino Gemelli IRCCS in Rome, a teaching hospital that oversees about 1500 MS patients. Inclusion criteria encompassed all patients who presented with a first neurological event suggestive of relapsing–remitting MS between January 2018 and July 2022 and who were successively diagnosed with MS according to the 2017 revision of McDonald’s criteria [70]. Data were collected from the review of medical records. For each patient, we evaluated the following characteristics at disease onset: smoking habit; body mass index (BMI); vitamin D levels; exposure to EBV; the presence, number, and localization of T2-FLAIR and gadolinium-enhancing lesions on magnetic resonance imaging (MRI); and the Expanded Disability Status Scale (EDSS) score. We also assessed clinical recovery after the first clinical event and after the first treatment started. Data on SARS-CoV-2 vaccination included the dates of administration of each dose and the type of vaccine (BNT162b2—Comirnaty© (BioNTech/Pfizer, Mainz, Germany/New York, NY, USA), mRNA-1273—Spikevax© (Moderna, Cmbridge MA, USA), ChAdOx1-S—Vaxzevria© (AstraZeneca PLC, Cambridge, UK), and Ad26.COV2.S—Johnson&Johnson©, Titusville, NJ, USA). For patients newly diagnosed with MS after 2020, we collected the time from vaccination to disease onset. For patients presenting with MS within 1 month from any dose of vaccine, we conducted genetic testing on blood samples, as well as in control groups as described below. Written consent was obtained from all patients. Anonymized data not published within this article are available upon request.

### 4.2. SARS-CoV-2 Vaccination

In Italy, the COVID-19 vaccination campaign started on 27 December 2020. The first vaccines available were mRNA vaccines and included BNT162b2–Comirnaty© (two intramuscular injections 21 days apart) [71] and mRNA-1273—Spikevax© (two intramuscular injections 28 days apart) [72]. Non-replicating vector vaccines, namely ChAdOx1-S—Vaxzevria© (two intramuscular injections 4–12 weeks apart) [73] and Ad26.COV2.S—Johnson&Johnson© (only one intramuscular injection) [74], were available from February 2021 and April 2021, respectively.

Age and pre-existing medical conditions as the main variables associated with mortality from COVID-19 guided the order of priority for vaccination starting with people aged over 80 and people at risk of developing severe COVID-19, including patients suffering from neurological conditions such as MS [75]. Other categories were also identified as priority groups, regardless of age and pre-existing medical conditions, such as caregivers of frail people, social and healthcare workers on the frontline of the COVID-19 response, school staff, etc. After that, vaccination was offered to the remaining population.

In April 2021, ChAdOx1-S and Ad26.CoV2.S. distribution was restricted to people older than 60 years due to some reports of cases of severe thromboembolism in younger people. Since then, the vaccination campaign has been carried out almost entirely through mRNA-based vaccines (European COVID-19 vaccine-safety-update; website last visited: 31 May 2023) [76].

From October 2021, a booster dose with one of the two m-RNA vaccines (original/omicron BA.1 or original/omicron BA.4–5) was offered to subjects who completed the primary series in order of priority. Since April 2022, a second booster dose has been gradually recommended to a wider target population once at least 120 days have elapsed since the last dose or the last SARS-CoV-2 infection.

In Italy, 145,134,032 vaccine doses have been administered until July 2022 (Italian Task Force COVID-19 update; website last visited: 13 March 2024) [77].

### 4.3. HLA-DR Genotyping

Genomic DNA for HLA-DR genotyping was extracted using QIAamp DNA Mini kits (Qiagen GmbH, Hilden, Germany); it was amplified in the region of exon 2 of HLA-DRB1 through PCR and then was reversely hybridized using the INNO-LiPA HLA-DRB1 plus kit (Fujirebio Italia, Pomezia, Italy), following the manufacturer’s instructions [77,78]. LiRAS software version 2.0 also provided by Fujirebio was used for the interpretation of HLA-DRB1 probes on the hybridization strips to predict one-digit HLA-DRB1 [79,80].

### 4.4. Statistical Analysis

We described the general characteristics of the population with summary statistics. We used the Mann–Whitney U-test and Pearson’s chi-square test/Fisher’s exact test as appropriate to test differences between groups. We used the Shapiro–Wilk test as the normality test. We evaluated univariate associations between patients diagnosed with MS during the pre-COVID-19 era and during the COVID-19 era.

To define a possible temporal association between SARS-CoV-2 vaccination and the onset of the acute demyelinating event, we established a one-month timeframe based on recent literature about inflammatory events related to SARS-CoV-2 vaccination [26]; a two-month timeframe was also investigated according to the classical relationship between demyelinating events and vaccination by Confavreux [81].

Since our patients were not randomized, we used propensity score matching (PSM) to estimate differences in outcome measures between patients diagnosed after SARS-CoV-2 vaccination and all other patients, considering a maximum of one month between vaccination and MS onset. Covariates included in this model were age, sex, smoke, positive EBV serology, vitamin D serum levels, and BMI (kg/m^2^). The “greedy nearest neighbor” matching method was used to find pairs of observations that have very similar propensity scores, setting a caliper of 0.02. The significance threshold was set at 5%. All analyses were performed using R software v.4.1.3 (https://www.r-project.org, accessed on 10 October 2023).

## 5. Conclusions

These data, although limited by the restricted number of patients, once again corroborate and confirm the idea that the etiopathogenesis of relapsing–remitting MS can be determined by a myriad of factors, of both genetic and environmental origins, whose interactions lead in some individuals to the development of the disease and, more generally, of autoimmune disorders [82,83,84,85,86]. Understanding these individual factors could have a great impact on managing patients and choosing the best treatment options. Our study also demonstrates the need for in-depth context-specific genetic and environmental data to carefully delineate the causal role and the potential evolution of each MS case individually.

## Figures and Tables

**Figure 1 ijms-25-04556-f001:**
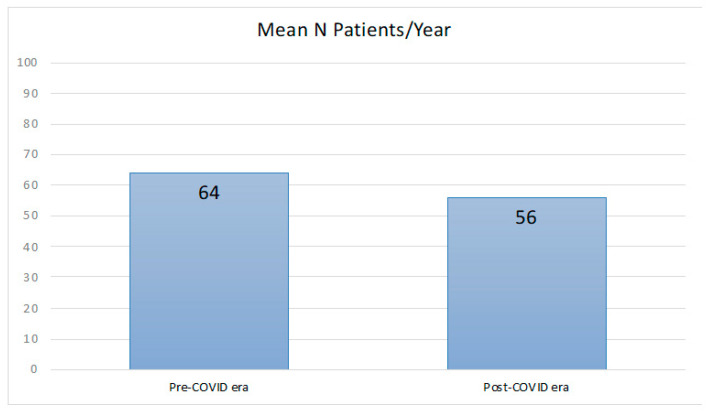
Mean number of new MS onset cases per year; Pre-COVID-19 era: January 2018–March 2020; and Post-COVID-19 era: April 2020–July 2022. A statistically significant difference was not found.

**Figure 2 ijms-25-04556-f002:**
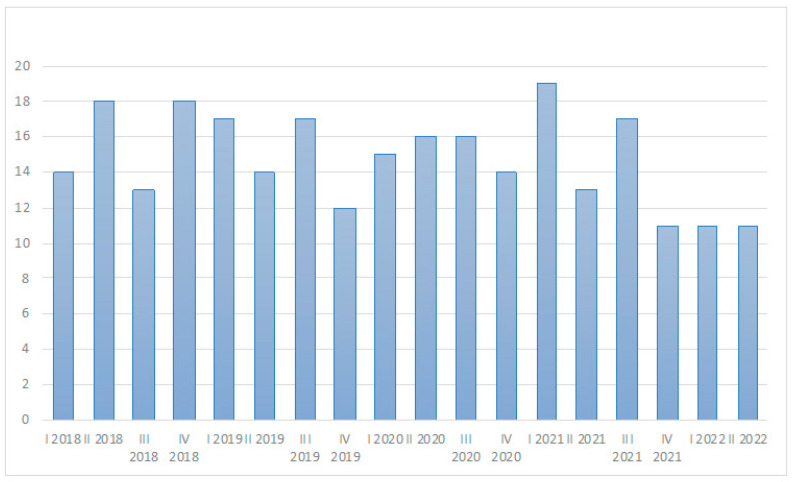
Number of new SM onset cases per trimester, January 2018 to July 2022; the Kruskal–Wallis test was applied, and no significant differences between groups have been found.

**Figure 3 ijms-25-04556-f003:**
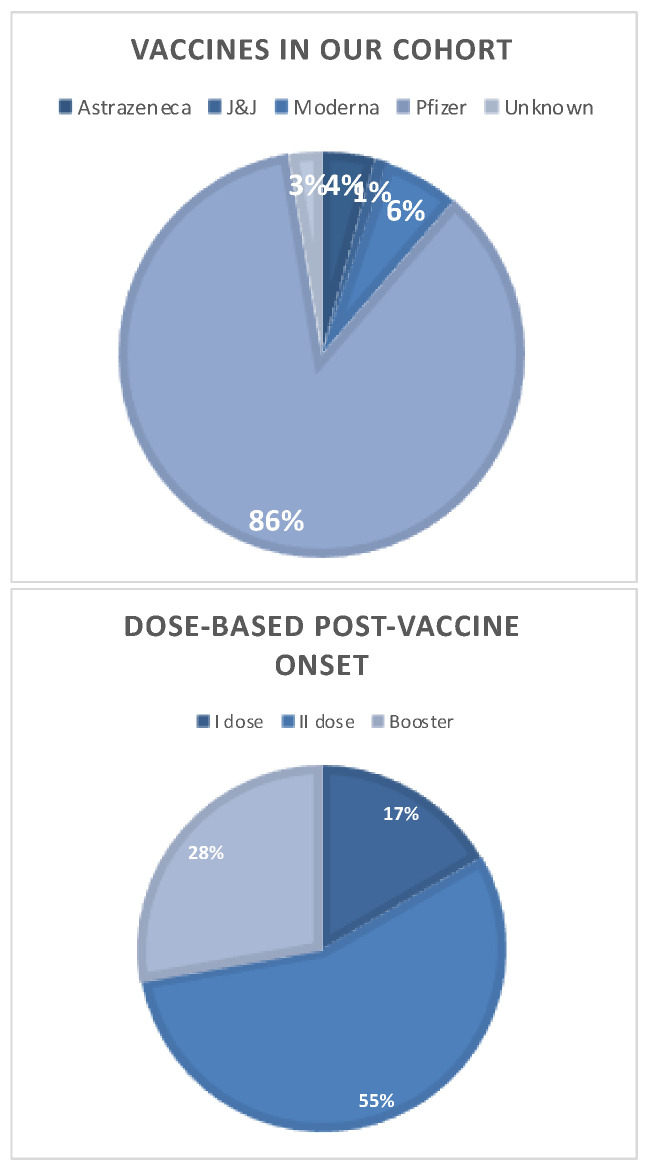
In the first diagram, the distribution of vaccines in our cohort (266 patients) is represented; in the second diagram, we represent the types of doses administered to patients who, after that, presented a post-vaccine MS onset.

**Figure 4 ijms-25-04556-f004:**
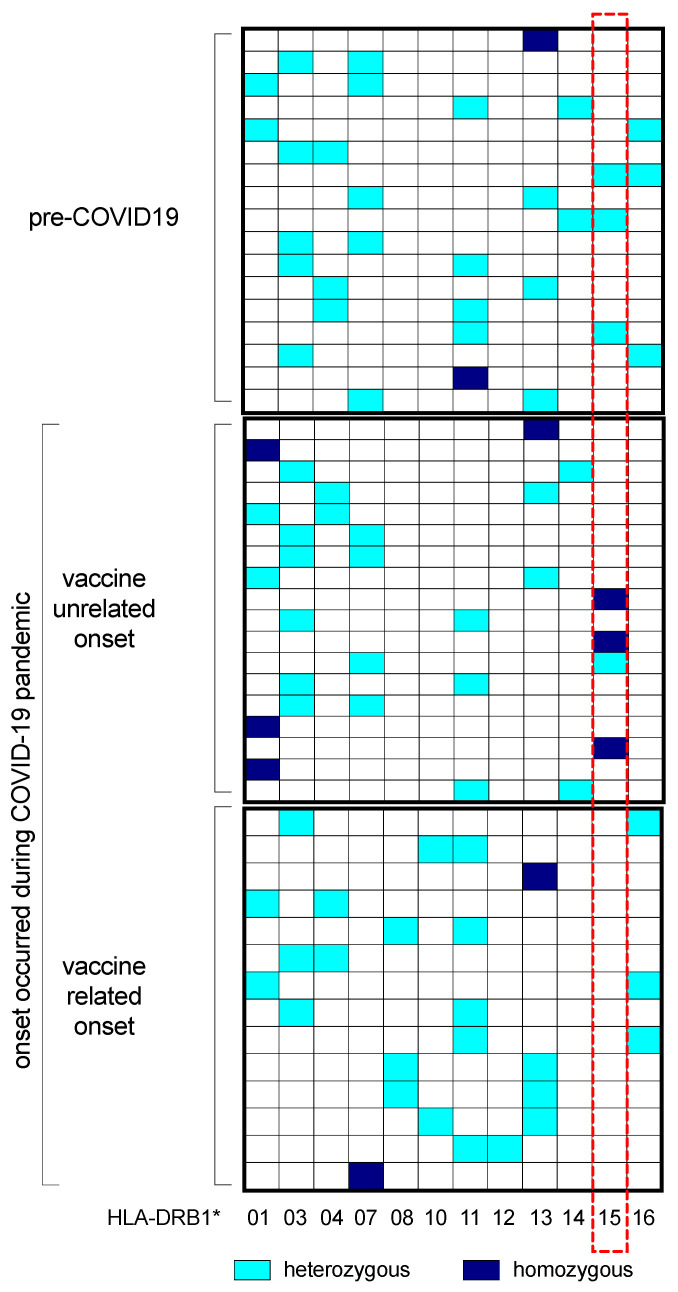
HLA-DRB1 haplotypes. Comparison of HLA-DRB1 haplotypes (both alleles) among 17 patients with disease onset in the pre-COVID-19 era (control group 1), 18 patients with disease onset in the post-COVID-19 era but not vaccine-related (control group 2), and 14 patients with a post-vaccine new diagnosis of MS (cases). Each line indicates a single patient. Positive HLA-DRB1 alleles with a square filled in light blue for heterozygous and dark blue for homozygous. In the red box, the column concerning HLA-DRB1*15 is not expressed in the subgroups of MS patients with disease onset related to vaccination.

**Table 1 ijms-25-04556-t001:** Comparison between patients with a new diagnosis of MS in the pre-COVID-19 era and during the COVID-19 era.

	Before COVID-19 Era	During COVID-19 Era	*p*-Value
Patients with a new diagnosis of MS	143	123	
Women (%)	105/143 (73.4%)	80/123 (65.0%)	0.178
Age in years—median (IQR)	33.0 (25.82–39.89)	33.0 (25.82–39.89)	0.052
Family history of MS (%)	8/143 (5.6%)	10/123 (8.1%)	0.565
Autoimmune comorbidity (%)	18/143 (12.6%)	7/123 (5.7%)	0.087
Time from onset to diagnosis—median (IQR)	7.0 (1.97–7.70)	4.0 (1.97–7.70)	<0.001
Type of onset			0.081
Optic neuritis (%)	29/143 (20.3%)	21/123 (17.1%)	
Brain stem (%)	31/143 (21.6%)	22/123 (17.9%)	
Cerebellum (%)	3/143 (2.1%)	1/123 (0.8%)	
Spinal cord (%)	38/143 (26.6%)	34/123 (27.6%)	
Supratentorial (%)	20/143 (14.0%)	10/123 (8.1%)	
Multisystemic (%)	22/143 (15.4%)	35/123 (28.5%)	
Oligoclonal bands (%)	122/139 (87.8%)	100/110 (90.9%)	0.528
EDSS at onset—median (IQR)	2.0 (1.50–2.5)	2.0 (1.50–2.5)	0.097
EDSS after relapse—median (IQR)	1.0 (1.50–2.5)	1.0 (1.50–2.5)	0.905
Clinical recovery after pulsed steroid therapy			0.372
None (%)	7/143 (4.9%)	10/123 (8.1%)	
Partial (%)	89/143 (62.2%)	74/123 (60.2%)	
Full (%)	48/143 (33.6%)	38/123 (30.9%)	
MRI brain stem (%)	60/143 (42.0%)	49/123 (39.8%)	0.822
MRI cerebellum	41/143 (28.7%)	35/123 (28.5%)	1.000
MRI spinal cord (%)	99/143 (69.2%)	93/123 (75.6%)	0.308
MRI infratentorial (%)	79/143 (55.2%)	61/123 (49.6%)	0.506
MRI number of supratentorial lesions—median (IQR)	5 (3.0–9.0)	5 (3.0–9.0)	0.215
MRI contrast enhancement (%)	76/143 (53.1%)	68/123 (55.3%)	0.637
Highly active efficacy therapy (%)	62/143 (43.4%)	60/114 (52.6%)	0.218
Smoke (%)	46/143 (32.2%)	41/123 (33.3%)	0.943
Vitamin D—median (IQR)	23.9 (17.55–30.76)	23.9 (17.55–30.76)	0.564
BMI (kg/m^2^)—median (IQR)	23.6 (21.30–26.23)	23.6 (21.30–26.23)	0.383
EBV positive serology (%)	124/143 (86.7%)	110/123 (89.4%)	0.065

**Table 2 ijms-25-04556-t002:** Comparison between patients with post-vaccine and non-vaccine-related new diagnosis of MS considering a maximum of 1 month between SARS-CoV-2 vaccination and demyelinating event onset.

	Post-Vaccine New Diagnosis of MS	Non-Vaccine-Related New Diagnosis of MS	*p*-Value
Number of patients with a new diagnosis of MS	14	252	
Women (%)	11/14 (78.6%)	174/252 (69.0%)	0.623
Age in years—median (IQR)	33.0 (26.00–40.00)	33.0 (26.00–40.00)	0.321
Family history of MS (%)	1/14 (7.1%)	16/252 (6.3%)	1.000
Autoimmune comorbidity (%)	0/14 (0.0%)	24/252 (9.5%)	0.456
Time from onset to diagnosis—median (IQR)	4.0 (1.97–7.70)	4.0 (2.00–8.00)	0.869
Type of onset			0.379
Optic neuritis (%)	0/14 (0.0%)	50/252 (19.8%)	
Brain stem (%)	5/14 (35.7%)	48/252 (19.0%)	
Cerebellum (%)	0/14 (0.0%)	4/252 (1.6%)	
Spinal cord (%)	4/14 (28.6%)	68/252 (27.0%)	
Supratentorial (%)	1/14 (7.1%)	29/252 (11.5%)	
Multisystemic (%)	4/14 (28.6%)	53/252 (21.0%)	
Oligoclonal bands (%)	10/12 (83.3%)	212/237 (89.5%)	0.863
EDSS at onset—median (IQR)	2.0 (1.50–2.50)	2.0 (1.50–2.50)	0.328
EDSS after relapse—median (IQR)	1.0 (0.00–1.50)	1.0 (0.00–1.50)	0.264
Clinical recovery after pulsed steroid therapy			0.180
None (%)	1/14 (7.1%)	14/252 (5.6%)	
Partial (%)	3/14 (21.4%)	151/252 (59.9%)	
Full (%)	4/14 (28.6%)	82/252 (32.5%)	
MRI brain stem (%)	6/14 (42.9%)	103/252 (40.9%)	1.000
MRI cerebellum (%)	5/14 (35.7%)	71/252 (28.2%)	0.688
MRI spinal cord (%)	10/14 (71.4%)	182/252 (72.2%)	1.000
MRI infratentorial (%)	9/14 (64.3%)	132/252 (52.4%)	0.517
MRI number of supratentorial lesions—median (IQR)	5.0 (3.00–9.00)	5.0 (3.00–9.00)	0.428
MRI contrast enhancement (%)	11/14 (78.6%)	131/252 (52.0%)	0.102
Highly active efficacy therapy (%)	8/10 (80.0%)	114/246 (46.3%)	0.073
Smoke (%)	4/14 (28.6%)	83/252 (32.9%)	0.931
Vitamin D—median (IQR)	23.9 (17.55–30.76)	23.9 (17.55–30.76)	0.431
BMI (kg/m^2^)—median (IQR)	23.6 (21.30–26.23)	23.6 (21.30–26.23)	0.334
EBV positive serology (%)	13/14 (92.9%)	205/252 (81.3%)	0.544

**Table 3 ijms-25-04556-t003:** Comparison among 17 patients with disease onset in the pre-COVID-19 era (control group 1), 18 patients with disease onset in the post-COVID-19 era but not vaccine-related (control group 2), and all 14 patients with a post-vaccine new diagnosis of MS (cases). Most relevant HLA-DRB1 alleles are displayed as allele frequencies. Sums of positives were not intended as a total due to the presence for each MS patient of two alleles that in a few cases are present in homozygosis.

HLA-DRB1 Alleles of Patients with a New Diagnosis of MS.			
HLA-DRB1 Alleles	Pre-COVID-19 Era	COVID-19 Era	
		Vaccine-Unrelated Onset	Vaccine-Related Onset
HLA-DRB1*01, *n* (%)	2.0 (11.8)	4.0 (22.2)	2.0 (14.3)
HLA-DRB1*03, *n* (%)	5.0 (29.4)	6.0 (33.3)	3.0 (21.4)
HLA-DRB1*04, *n* (%)	3.0 (17.6)	2.0 (11.1)	2.0 (14.3)
HLA-DRB1*07, *n* (%)	5.0 (29.4)	4.0 (22.2)	2.0 (14.3)
HLA-DRB1*11, *n* (%)	5.0 (29.4)	3.0 (16.7)	2.0 (14.3)
HLA-DRB1*13, *n* (%)	4.0 (23.5)	3.0 (16.7)	5 (35.7)
HLA-DRB1*15, *n* (%)	3.0 (17.6)	4.0 (22.2)	0

## Data Availability

The datasets analysed during the current study are available from the corresponding author upon reasonable request.

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
