# Peer review of "Multiple Sclerosis Onset before and after COVID-19 Vaccination: Can HLA Haplotype Be Determinant?"

_ijms, 2024, doi:10.3390/ijms25084556_

Round 1

Reviewer 1 Report

Comments and Suggestions for Authors

Thank you for providing me the opportunity to review this paper. In this study authors present the pattern of HLA gentyping in patients with MS after COVID-19 vaccination. The topic is highly relevant as many patients are asking about the possibility of worsening or developing MS after vaccines. Although changes in HLA pattern after the vaccination could indicate that there are some changes occur in immunological system of MS patients, this could be a positive not negative finding. Could it not be a manifestation to development of immune response to vaccine? It would be crucial to compare this with healthy individuals and this should be mentioned as a limitation of the study. Generally, the limitation section should be introduced in the paper and discussed. Maybe it skipped my attention but was also the pattern of immunomodulatory treatment discussed between the typical MS cohort and post COVID-19 vacinnation cohort? This can have impact on results. Also I would suggest to include in the discussion these important studies about immunological response in patients with MS after COVID-19 vaccine: Safety of Vaccines against SARS-CoV-2 among Polish Patients with Multiple Sclerosis Treated with Disease-Modifying Therapies - PubMed (nih.gov), Analysis of seroconversion following COVID-19 vaccination among multiple sclerosis patients treated with disease-modifying therapies in Poland - PubMed (nih.gov)

Reviewer 2 Report

Comments and Suggestions for Authors

This study aimed to compare cases of newly diagnosed MS reported before and after the outbreak of the COVID-19 pandemic and their temporal relationship with SARS-CoV-2 vaccination. The topic is clinically interesting, but several issues require clarification. Below are my comments:

1.       The title of the study deviates from its main purpose.

2.       The percentage of patients with spinal cord disease onset is surprisingly high.

3.       Only patients with the relapsing-remitting form of the disease were included in the study, which should be more emphasized.

4.       HLA Genotyping was performed on plasma samples collected only from 49 MS patients, which significantly limits the interpretation of the study results and is its limitation.

5.       The limitations of the study are too sparsely described.

Reviewer 3 Report

Comments and Suggestions for Authors

The authors in the present study aimed to evaluate any connections between SARS-CoV2 vaccination and MS onset and to find any link between the disease onset and the vaccination in clinical, MRI presentation or even on the genetic level. 

The main finding of the study, that different HLA alleles were present in the vaccine-related and unrelated group is interesting and highlights the need for more in depth assessment on the matter, namely the re-assessing of MS pathogenesis on the genetic level.

The introduction is informative, the patient and methods section is adequate and the statistical analysis is well described. Results are clearly presented and the Discussion is good and not too far-reaching regarding the conclusions.

All in all, I advise the manuscript to be accepted.
